# QUANTILE ADVANTAGE ESTIMATION: STABILIZING RLVR FOR LLM REASONING

**Junkang Wu**[1]  **Kexin Huang**[1]  **Jiancan Wu**[1*]**An Zhang**[1]
**Xiang Wang**[1*]**Xiangnan He**[2]
[1]University of Science and Technology of China
[2]MoE Key Lab of BIPC, University of Science and Technology of China
{jkwu0909, wujcan, xiangnanhe}@gmail.com

## ABSTRACT

Reinforcement Learning with Verifiable Rewards (RLVR) strengthens LLM reasoning but training often oscillates between entropy collapse and entropy explosion. We trace both hazards to the mean-baseline used in value-free RL (*e.g.,* GRPO & DAPO), which improperly penalizes negative-advantage samples under reward outliers. We propose Quantile Advantage Estimation (QAE), replacing the mean with a group-wise $K$-quantile baseline. QAE induces a response-level, two-regime gate: on hard queries ($p \leq 1-K$) it reinforces rare successes, while on easy queries ($p > 1-K$) it targets remaining failures. Under first-order softmax updates, we prove two-sided entropy safety, giving lower/upper bounds on one-step entropy change that curb explosion and prevent collapse. Empirically, this minimal modification stabilizes entropy, sparsifies credit assignment (with tuned $K$, roughly 80% of responses receive zero advantage), and yields sustained pass@1 gains on across AIME'24/'25 and AMC'23. These results identify baseline design—rather than token-level heuristics—as the primary mechanism for scaling RLVR [1].

## 1 INTRODUCTION

Reinforcement Learning with Verifiable Rewards (RLVR) (Lambert et al., 2024; DeepSeek-AI et al., 2025; Yang et al., 2025a) enhances Large Language Models (LLMs) by rewarding verifiable correctness (Phan et al., 2025; Rein et al., 2023). Yet reward-driven optimization often triggers *entropy collapse* (Yu et al., 2025; Cui et al., 2025): the policy distribution sharpens prematurely, suppressing exploration and ultimately limiting performance. This exposes a fundamental tension between maximizing reward and preserving policy diversity during RLVR fine-tuning.

Prior work focuses almost exclusively on preventing collapse, *e.g.,* uplifting low-probability tokens (Yu et al., 2025), penalizing collapse-inducing tokens (Cui et al., 2025), or preserving policy diversity by primarily learning from negative samples (Zhu et al., 2025). While effective at avoiding collapse, these methods address only one side of the problem and largely overlook its symmetric counterpart: *entropy explosion*. Uncontrolled entropy growth is equally harmful, leading to inefficient exploration and stalled progress.

This risk is practical, not merely theoretical. On Qwen3-8B-Base with DAPO, Figure 1 (left) shows that `Clip-Higher` averts collapse but induces an early entropy spike (steps $10 \rightarrow 80$) that, while not immediately harming performance, creates long-term instability. After step 100, entropy remains high and volatile, while performance plateaus. These dynamics highlight key shortcomings of unconstrained entropy growth: (i) higher policy entropy does not guarantee continued effective exploration—performance can plateau despite ongoing behavioral variability reflected in high entropy; and (ii) the initial entropy spike indicates a period of over-exploration that, though not immediately destructive, ultimately undermines the model's ability to consolidate learning from high-reward reasoning trajectories. The dual challenge, therefore, is to avoid both premature convergence (collapse)

---

*Jiancan Wu and Xiang Wang are the corresponding authors.
[1]The code is available at https://github.com/junkangwu/QAE.

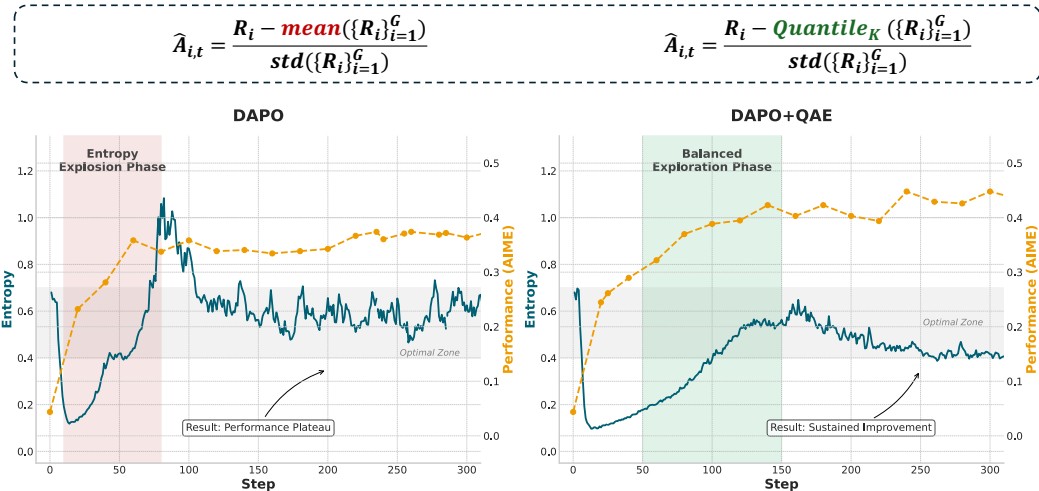

Figure 1: Entropy–performance dynamics on Qwen3-8B-Base. Left: DAPO with `Clip-Higher` prevents early collapse but triggers an early entropy spike (steps 10–80) and a later performance plateau. Right: our quantile baseline (QAE) stabilizes policy entropy and sustains pass@1 gains by steering training into a balanced exploration regime.

and unproductive, signal-degrading divergence (explosion). Merely avoiding collapse is therefore insufficient—effective RLVR requires keeping entropy within a productive range.

We address this dual challenge with **Quantile Advantage Estimation (QAE)**, which dynamically regulates policy entropy by replacing the conventional mean reward baseline with a group-wise $K$-quantile. The key idea is that the baseline choice controls how many samples receive positive vs. negative advantages, which directly impacts exploration behavior. Specifically, a lower $K$ marks more samples as having positive advantage, encouraging the model to exploit these successful patterns and reducing entropy. Conversely, a higher $K$ makes fewer samples appear successful, pushing the model to diversify its behavior patterns, thereby increasing entropy. By tuning the quantile parameter $K$, we can control the exploration-exploitation balance. As shown in Figure 1 (right), with an appropriately chosen $K$, this mechanism steers training toward a stable entropy regime — neither collapsing nor exploding — enabling sustained performance gains beyond the prior plateau. This mechanism has a striking empirical consequence: **it naturally sparsifies updates**. With a tuned $K$, roughly 80% of responses receive zero advantage. This concentrates computational effort on the most informative samples and revealing a deep redundancy in standard mean-baseline approaches.

We trace both early entropy spikes and late plateaus to the mean-baseline in value-free RL; substituting a $K$-quantile baseline (QAE) implements a response-level gate that routes updates to rare successes on hard queries and to remaining failures on easy ones. We prove a two-sided entropy safety guarantee and derive a discriminative objective that explains the observed stability, which leads to significant pass@1 gains and solid pass@16 performance. Empirically, the one-line swap boosts `Clip-Higher` (Yu et al., 2025) on QWEN3-8B/14B-BASE, pairs well with `Clip-Cov`/`KL-Cov` (Cui et al., 2025) on QWEN3-8B-BASE, and works with `GSPO` (Zheng et al., 2025) on QWEN3-30B-A3B-BASE, yielding consistent pass@1 gains and strong pass@16 on AIME'24, AIME'25, and AMC'23. Overall, QAE reframes entropy regulation as a **baseline-design** problem rather than a **token-level** tuning problem.

## 2 PRELIMINARIES

In this section, we review the policy optimization algorithms that form the foundation of our work, starting with Proximal Policy Optimization (PPO) and its value-free variants, GRPO and DAPO.

**Proximal Policy Optimization (PPO)** PPO (Schulman et al., 2017) is a foundational on-policy algorithm that stabilizes training by constraining policy updates to a trust region around the previous

policy $\pi_{\boldsymbol{\theta}_{\text{old}}}$. It maximizes a clipped surrogate objective:

$$\mathcal{J}_{\text{PPO}}(\theta) = \mathbb{E}_{(\boldsymbol{q},\boldsymbol{a})\sim\mathcal{D},\boldsymbol{o}\sim\pi_{\boldsymbol{\theta}_{\text{old}}}(\cdot|\boldsymbol{q})} \left[ \min\left( r_t(\theta)\hat{A}_t, \, \text{clip}(r_t(\theta), 1-\epsilon, 1+\epsilon)\hat{A}_t \right) \right], \quad (1)$$

where $r_t(\boldsymbol{\theta}) = \frac{\pi_{\boldsymbol{\theta}}(o_t|\boldsymbol{q},\boldsymbol{o}_{<t})}{\pi_{\boldsymbol{\theta}_{\text{old}}}(o_t|\boldsymbol{q},\boldsymbol{o}_{<t})}$ is the probability ratio. The advantage $\hat{A}_t$ is typically estimated by a value network, and $\epsilon$ is the clipping hyperparameter (*e.g.,* 0.2).

**Group Relative Policy Optimization (GRPO)**   To eliminate the need for a value network, GRPO (Shao et al., 2024) adapts the PPO objective by proposing a relative advantage estimator. For each query, GRPO samples a group of $G$ responses $\{\boldsymbol{o}_i\}_{i=1}^{G}$ from $\pi_{\boldsymbol{\theta}_{\text{old}}}$. Each response is assigned a binary reward $R_i$ based on its correctness against a ground-truth answer $\boldsymbol{a}$. The advantage for the $i$-th sample is then estimated by normalizing its reward against the group's statistics:

$$\hat{A}_i = \frac{R_i - \text{mean}(\{R_k\}_{k=1}^{G})}{\text{std}(\{R_k\}_{k=1}^{G})}, \quad \text{where} \ \ R_i = \begin{cases} 1.0 & \text{if } \texttt{is\_equivalent}(\boldsymbol{a},\boldsymbol{o}_i), \\ 0.0 & \text{otherwise.} \end{cases} \quad (2)$$

GRPO further incorporates a KL divergence penalty against  to regularize the policy update.

**Dynamic Sampling Policy Optimization (DAPO)**   We use DAPO (Yu et al., 2025), a state-of-the-art value-free method, as our baseline. DAPO refines GRPO with several key modifications. It removes the KL penalty but introduces an asymmetric clipping range $(1 - \epsilon_{\text{low}}, 1 + \epsilon_{\text{high}})$, allowing larger updates for advantageous actions. The objective is also normalized at the token level:

$$\mathcal{J}_{\text{DAPO}}(\theta) = \mathbb{E}_{\substack{(\boldsymbol{q},\boldsymbol{a})\sim\mathcal{D}, \\ \{\boldsymbol{o}_i\}_{i=1}^{G}\sim\pi_{\boldsymbol{\theta}_{\text{old}}}(\cdot|\boldsymbol{q})}} \left[ \frac{1}{Z} \sum_{i=1}^{G} \sum_{t=1}^{|\boldsymbol{o}_i|} \min\left( r_{i,t}(\theta)\hat{A}_{i,t}, \, \text{clip}\big(r_{i,t}(\theta), 1-\epsilon_{\text{low}}, 1+\epsilon_{\text{high}}\big)\hat{A}_{i,t} \right) \right]$$

where $Z = \sum_{i=1}^{G} |\boldsymbol{o}_i|$ is the total number of tokens in the group, and the advantage $\hat{A}_{t,i}$ is computed as in GRPO. Crucially, DAPO employs a dynamic sampling constraint:

$$0 < |\{\boldsymbol{o}_i \mid \texttt{is\_equivalent}(\boldsymbol{a},\boldsymbol{o}_i)\}| < G.$$

This ensures that each training batch contains both positive and negative examples, guaranteeing a meaningful advantage signal and stable gradients.

# 3   THE ENTROPY DILEMMA IN RL SCALING: FROM COLLAPSE TO EXPLOSION

Policy entropy is central to reinforcement learning, governing the exploration–exploitation trade-off. This balance is especially fragile in RLVR for large models. When entropy is too low, the policy converges prematurely to suboptimal behaviors (*entropy collapse*); when it is too high, uncontrolled stochasticity attenuates learning signals (*entropy explosion*). Navigating this entropy dilemma is therefore pivotal for scaling RLVR.

## 3.1   THE TWO PERILS OF POLICY ENTROPY

**Entropy collapse.**   Well documented in RLVR (Yu et al., 2025; Cui et al., 2025; Zhu et al., 2025), collapse occurs when the policy becomes overly deterministic too early. The resulting loss of exploration traps training in narrow reasoning modes and limits generalization.

**Entropy explosion.**   At the other extreme, the policy becomes overly stochastic: gradients are swamped by noise, credit assignment deteriorates, and learning turns unstable and inefficient—an equally limiting regime that has been comparatively underexplored (Ahmed et al., 2019; Geist et al., 2019; Haarnoja et al., 2018; Xu et al., 2021; Zhang et al., 2025).

**The dilemma.**   Most prior work targets collapse alone. Treating it as the sole bottleneck is a critical oversight: in practice, mitigating collapse with existing techniques can inadvertently induce explosion. Addressing only one side is insufficient; effective RLVR requires keeping policy entropy within a productive, stable range. We next analyze the mechanisms that drive entropy explosion and motivate our remedy.

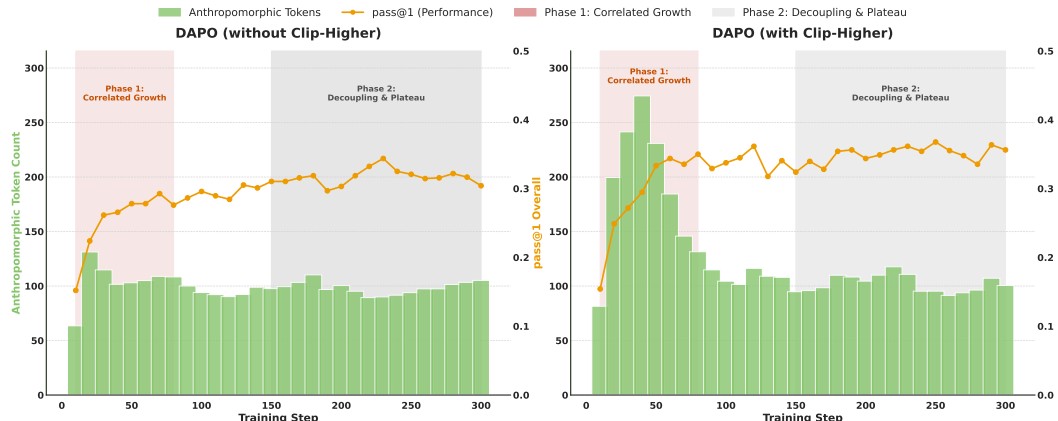

Figure 2: **DAPO training dynamics on Qwen3–8B.** *Left*: without `Clip-Higher`; *Right*: with `Clip-Higher`. In both settings we observe two phases—an early *correlated growth* between anthropomorphic token frequency and pass@1, followed by a *decoupling then plateau*. While `Clip-Higher` averts collapse, it does not prevent the later performance stall.

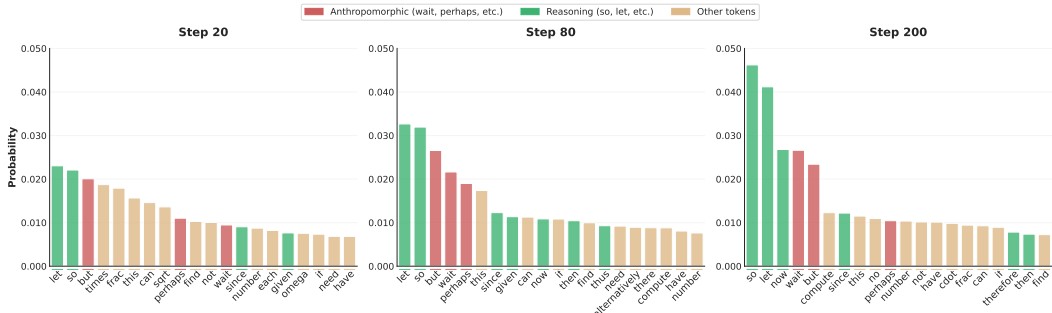

Figure 3: **Evolution of high-entropy token usage under DAPO (steps 20/80/200).** Early training exhibits diverse anthropomorphic tokens (*e.g., wait*, *perhaps*); by steps 80–200 the distribution homogenizes around rigid reasoning templates (*e.g., so*, *let*), indicating reduced exploratory diversity consistent with entropy explosion.

## 3.2 AN ANALYSIS OF ENTROPY EXPLOSION IN RLVR

To investigate the drivers of entropy explosion, we analyze a prevalent class of value-free RL methods that apply policy gradients at the *token level*. We use DAPO (Yu et al., 2025) as a representative case, focusing on its `Clip-Higher` mechanism—a token-level control designed to prevent entropy collapse but, as we will show, one that also illustrates the pitfalls of fine-grained control. Unless otherwise noted, we follow the recommended configurations in Yu et al. (2025); full details appear in Appendix B.1.

**Observation 1: Token-level control does not guarantee sustained reasoning gains.** In Figure 2, `Clip-Higher` triggers an early spike (steps 20–80) in anthropomorphic tokens—proposed by Yang et al. (2025b) as markers of "aha-moment" reasoning—that coincides with sharp pass@1 gains. However, after step 150, anthropomorphic token frequency returns toward baseline while performance plateaus.

**Observation 2: Token-level control yields homogenized, low-quality exploration.** To probe the stall, we examine the distribution of high-entropy tokens at steps 20, 80, and 200 (*cf.* Figure 3). Early in training, diverse markers such as *wait* and *perhaps* are frequent. By step 80, usage concentrates on assertive, formulaic tokens like *so* and *let*. This convergence reflects a loss of diversity in high-entropy states: the model increasingly relies on rigid reasoning templates rather than exploring alternatives, aligning with the observed plateau.

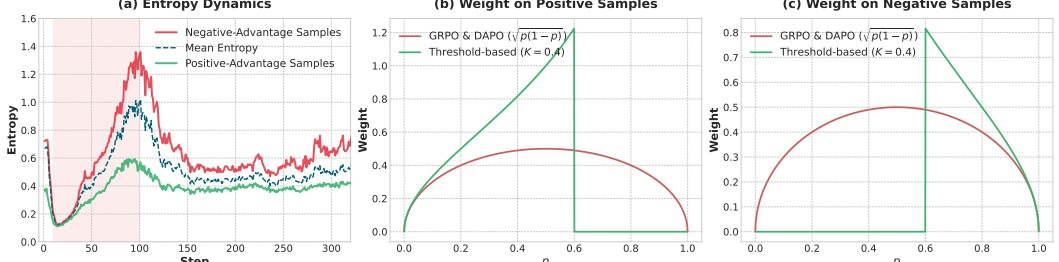

Figure 4: **Quantile baseline reshapes weighting and entropy dynamics.** *Left*: policy entropy over training split by advantage sign—negative-advantage samples drive the surge. *Middle/Right*: query-level weights vs. success rate $p$; GRPO & DAPO use symmetric $\sqrt{p(1-p)}$ weighting, whereas our method applies a thresholded scheme ($K=0.4$).

Table 1: Different $\epsilon_{\text{high}}$ values in DAPO.

| $\epsilon_{\text{high}}$ | **AIME24** |
|---|---|
| 0.20 | $32.29^{-18.6\%}$ |
| 0.22 | $34.90^{-12.1\%}$ |
| 0.24 | $34.17^{-13.9\%}$ |
| 0.26 | $40.63^{+2.4\%}$ |
| 0.28 | $39.69$ |

**Observation 3: Entropy explosion is disproportionately driven by negative-advantage samples.** We decompose entropy dynamics by sample advantage, where positive-advantage samples contribute positive updates and negative-advantage samples contribute non-positive updates. As shown in Figure 4 (Left), entropy growth is dominated by negative-advantage samples, which show both the steepest increase and the largest share of entropy early in training. Positive-advantage samples remain comparatively stable. This imbalance indicates over-exploration induced by negative-advantage samples in the early phase, followed by insufficient exploitation later.

**Observation 4: Tuning token-level hyperparameters is insufficient.** One might lower the token-level high clip threshold $\epsilon_{\text{high}}$ to curb update magnitude. Table 1 (varying $\epsilon_{\text{high}}$ from 0.20 to 0.28) shows only marginal effects: performance peaks near $\epsilon_{\text{high}} = 0.26$, but the overall improvement is limited and the late-stage plateau persists. Simply adjusting token-level clipping cannot resolve the core exploration–exploitation tension.

> **TAKEAWAY**
>
> Our analysis indicates that fine-grained, token-level controls provide a temporary fix with notable side effects:
>
> - They prevent **entropy collapse** but can inadvertently induce a performance-limiting **entropy explosion**.
> - The explosion is mechanically rooted in the **advantage baseline**, which systematically mishandles **negative-advantage samples** under reward outliers.
> - The issue is therefore a **baseline-design flaw**, not a hyperparameter tuning problem at the token level.

## 4 METHOD: QUANTILE-BASED ADVANTAGE ESTIMATION FOR ENTROPY REGULATION

Building on the analysis in Section 3, we identify the *advantage baseline* as the primary source of instability in RLVR. Value-free methods such as GRPO (Shao et al., 2024) and DAPO (Yu et al., 2025) use an empirical *mean* baseline that is sensitive to reward outliers: a few high-reward samples can inflate the baseline, turning otherwise competent responses into negative-advantage examples and penalizing useful exploration, which induces entropy collapse.

We address this by *quantile-based advantage estimation*. Replacing the mean with a distributional quantile yields a baseline that is (i) statistically robust and (ii) explicitly *controllable*. A single hyperparameter $K \in (0, 1)$ shifts the update focus between exploration and exploitation.

### 4.1 FORMULATION AND INTUITION

For a query $q$, sample $G$ responses $\{(o_i, R_i)\}_{i=1}^G$ with $o_i \sim \pi_{\text{old}}(\cdot \mid q)$ and binary rewards $R_i \in \{0, 1\}$. Let

$$p(q) := \frac{1}{G} \sum_{i=1}^G R_i$$

be the empirical success rate under $\pi_{\text{old}}$. Define the group empirical CDF

$$\widehat{F}_q(x) := \frac{1}{G} \sum_{j=1}^G \mathbf{1}\{R_j \leq x\},$$

and the (right-continuous) $K$-quantile baseline

$$b_K(q) := \mathsf{Q}_K(\{R_j\}_{j=1}^G) = \inf\{x : \widehat{F}_q(x) \geq K\}, \qquad K \in (0, 1).$$

We then define the standardized advantage

$$\hat{A}_i = \frac{R_i - b_K(q)}{\text{std}(\{R_j\}_{j=1}^G) + \varepsilon}, \qquad \varepsilon > 0, \tag{3}$$

where $\varepsilon$ prevents division by zero when $p \in \{0, 1\}$. For binary rewards, the baseline reduces to a threshold on $p(q)$:

$$b_K(q) = \begin{cases} 0, & p(q) \leq 1-K, \\ 1, & p(q) > 1-K. \end{cases} \tag{4}$$

This yields two regimes governed by the difficulty threshold $1-K$:

- **Hard (exploitation-focused),** $p(q) \leq 1-K$**.** The baseline is $0$. Incorrect responses ($R = 0$) have $\hat{A} = 0$, while rare correct responses ($R = 1$) receive $\hat{A} > 0$, reinforcing nascent successful trajectories.

- **Easy (exploration-focused),** $p(q) > 1-K$**.** The baseline is $1$. Correct responses have $\hat{A} = 0$, while remaining failures ($R = 0$) yield $\hat{A} < 0$, discouraging residual failure modes on already-solved queries.

Hence $K$ acts as a direct lever that regulates policy entropy by switching updates between rare successes (hard) and remaining failures (easy).

### 4.2 GRADIENT ANALYSIS

We adopt the discriminative perspective of GRPO introduced by DisCO (Li et al., 2025), which separates a query-level weight from a discriminative term. Let $\pi_{\text{old}}^+(\cdot \mid q)$ and $\pi_{\text{old}}^-(\cdot \mid q)$ denote the conditional distributions of responses with rewards $1$ and $0$, respectively. For a response $o$, let $s_\theta^+(o, q)$ and $s_\theta^-(o, q)$ denote score functions based on token-normalized policy ratios for positive/negative examples (see Appendix A.2 for exact forms).

**GRPO revisited.** Li et al. (2025) show that the GRPO objective can be written as

$$\mathcal{J}_{\text{GRPO}}(\theta) = \mathbb{E}_q\Big[ \underbrace{\sqrt{p(q)\big(1 - p(q)\big)}}_{\text{query weight}} \cdot \underbrace{\mathbb{E}_{o \sim \pi_{\text{old}}^+,\, o' \sim \pi_{\text{old}}^-}\big[s_\theta^+(o, q) - s_\theta^-(o', q)\big]}_{\text{discriminative term}} \Big], \tag{5}$$

with a symmetric weight that down-weights both very easy and very hard queries (*cf.* Fig. 4).

**Quantile-based objective.** Under Eqs. 3–4, the standardized advantage is non-zero on *only one* outcome type per regime. Substituting into a GRPO-style objective yields:

**Proposition 4.1** (Quantile-regulated objective)**.** *Assume binary rewards, group size $G \geq 2$, and the right-continuous empirical quantile. Using the standardized advantage in Eqs. 3–4, the learning objective is (up to a constant factor depending on $\varepsilon$) equivalent to*

$$\mathcal{J}_{\text{Quantile}}(\theta) = \mathbb{E}_q\Big[ \mathbf{1}\{p(q) \leq 1-K\} \sqrt{\tfrac{p(q)}{1-p(q)}}\, \mathbb{E}_{o \sim \pi_{\text{old}}^+(\cdot|q)} s_\theta^+(o, q)$$

$$- \mathbf{1}\{p(q) > 1-K\} \sqrt{\tfrac{1-p(q)}{p(q)}}\, \mathbb{E}_{o' \sim \pi_{\text{old}}^-(\cdot|q)} s_\theta^-(o', q)\Big]. \tag{6}$$

**Remark.** Please check Appendix A for all proofs. Compared to the GRPO objective in Eq. 5, QAE makes two crucial changes: (i) it selectively nullifies one of the discriminative terms based on query difficulty, and (ii) it replaces the symmetric, bell-shaped weight $\sqrt{p(1-p)}$ with asymmetric, monotonic factors—either $\sqrt{p/(1-p)}$ for hard queries or $\sqrt{(1-p)/p}$ for easy queries. This transforms the update mechanism from focusing on moderately difficult problems to amplifying signals from rare successes or residual failures (*cf.* Fig. 4).

### 4.3 THEORETICAL ANALYSIS: TWO-REGIME ENTROPY SAFETY

**Setup.** Adopt a bandit reduction in which producing a full response $y$ to $q$ is a single action. Let $\pi(\cdot \mid q)$ be the current softmax policy and $H(q)$ the token-averaged (length-normalized) policy entropy. Let $\widehat{A}$ denote the GRPO/DAPO-style token-normalized advantage (Sec. 4.2); more generally, write $A_b(y,q) = r(y,q) - b(q)$ for the response-level advantage with baseline $b(q)$. For binary rewards with group success rate $p(q)$, we use the right-continuous $K$-quantile baseline $b_K(q)$ (Eq. 4), *i.e.,* $b_K(q) = 0$ if $p(q) \leq 1-K$ and 1 otherwise. Under first-order logit updates of a softmax policy with step size $\eta > 0$, the entropy–covariance identity (adapted from Cui et al. (2025)) yields,

$$\Delta H(q) \approx -\eta \operatorname{Cov}_{y \sim \pi(\cdot|q)}\big(\log \pi(y \mid q), \pi(y \mid q) A_b(y,q)\big), \quad \eta > 0.$$

**Baseline as a linear knob.** For $b \in [0,1]$, define $F_q(b) := \operatorname{Cov}_\pi\big(\log \pi, \pi(r-b)\big)$ for $r \in \{0,1\}$. By linearity,

$$F_q(b) = F_q(0) - b \operatorname{Cov}_\pi(\log \pi, \pi), \qquad \operatorname{Cov}_\pi(\log \pi, \pi) > 0$$

whenever $\pi(\cdot \mid q)$ is non-uniform. Hence $\Delta H(q;b) = -\eta F_q(b)$ is strictly increasing in $b \in [0,1]$.

**Proposition 4.2** (Two-regime entropy safety of $K$-quantile). *Fix $q$ and a non-uniform $\pi(\cdot \mid q)$. Then:*

1. ***Low-success (explosion-proof).*** *If $p(q) \leq 1-K$ so $b_K(q) = 0$, then for any baseline $b \in [0,1]$ (including the mean $b = p(q)$ or token-level clipping/KL that keep $b$ unchanged),*

$$\Delta H(q;b_K) \leq \Delta H(q;b).$$

2. ***High-success (collapse-proof).*** *If $p(q) > 1-K$ so $b_K(q) = 1$, then for any $b \in [0,1]$,*

$$\Delta H(q;b_K) \geq \Delta H(q;b).$$

**Sequences vs. token-level controls.** Existing token-level controls are *one-sided*: they rescale step sizes but leave the response-level baseline $b(q)$ unchanged, so they cannot prevent explosion driven by negative-advantage samples. In contrast, the $K$-quantile baseline is *two-sided* (Prop. 4.2): $b_K = 0$ when $p(q) \leq 1-K$ (explosion-proof) and $b_K = 1$ when $p(q) > 1-K$ (collapse-proof), matching the two training regimes in Fig. 4.

> 🏷 **TAKEAWAY**
>
> **Method takeaways (QAE).**
>
> - $K$**-quantile as a response-level gate.** A single parameter $K$ yields a deterministic switch (Eqs. 3–4): hard queries ($p(q) \leq 1-K$) update on *rare successes* only; easy queries ($p(q) > 1-K$) update on *remaining failures* only (Fig. 4).
>
> - **Two-sided entropy safety (provable).** Under first-order softmax updates, the $K$-quantile baseline attains the *extremal* one-step entropy shift—minimal at $p(q) \leq 1-K$ (prevents explosion) and maximal at $p(q) > 1-K$ (prevents collapse); see Prop. 4.2.
>
> *Note:* Token-level mechanisms only rescale steps and do not change the response-level baseline, so they cannot realize these guarantees.

## 5 EXPERIMENTS

**Evaluation protocol.** We evaluate on three standard math–reasoning benchmarks: **AIME'24**, **AIME'25**, and **AMC'23**. All evaluations are *zero-shot*. For each query we sample $k = 32$ completions with temperature $T = 0.7$. We report pass@1 and pass@16 as accuracy metrics, together with

Table 2: Overall performance on the AIME'24/'25 and AMC'23 benchmarks. Our drop-in QAE consistently improves pass@1 across different models and methods, while maintaining comparable pass@16 scores. Red denotes an improvement and blue a decline.

| Model | Method | AIME25 | | AIME24 | | AMC23 | |
|---|---|---|---|---|---|---|---|
| | | Pass@1 | Pass@16 | Pass@1 | Pass@16 | Pass@1 | Pass@16 |
| Qwen3-8B-Base | Clip-Higher | 32.71 | 56.66 | 39.69 | 71.23 | 92.11 | 97.50 |
| | + QAE | $34.90^{+6.7\%}$ | $57.92^{+2.2\%}$ | $48.23^{+21.5\%}$ | $71.63^{+0.6\%}$ | $92.97^{+0.9\%}$ | $97.50^{+0.0\%}$ |
| | CLIP-Cov | 33.02 | 52.27 | 42.40 | 68.58 | 87.42 | 96.25 |
| | + QAE | $37.40^{+13.3\%}$ | $56.29^{+7.7\%}$ | $46.04^{+8.6\%}$ | $73.16^{+6.7\%}$ | $90.23^{+3.2\%}$ | $96.25^{+0.0\%}$ |
| | KL-Cov | 33.33 | 45.86 | 44.90 | 73.00 | 86.02 | 95.00 |
| | + QAE | $33.44^{+0.3\%}$ | $51.62^{+12.6\%}$ | $44.69^{-0.5\%}$ | $77.08^{+5.6\%}$ | $87.97^{+2.3\%}$ | $96.25^{+1.3\%}$ |
| Qwen3-30B-A3B-Base | GSPO | 31.15 | 46.59 | 43.75 | 67.91 | 90.00 | 99.39 |
| | + QAE | $32.50^{+4.3\%}$ | $48.01^{+3.0\%}$ | $47.50^{+8.6\%}$ | $71.72^{+5.6\%}$ | $89.38^{-0.7\%}$ | $97.21^{-2.2\%}$ |

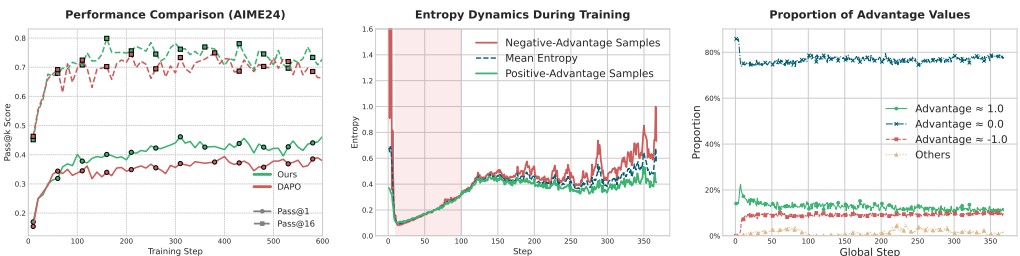

Figure 5: **Training dynamics and sparsity. (a)** AIME'24 (Qwen3–8B): QAE boosts pass@1 while keeping pass@16 comparable—showing higher sample efficiency. **(b)** Entropy by sign: DAPO's explosion stems from negative-advantage samples; QAE suppresses it. **(c)** Response sparsity: 80% responses have zero advantage, focusing updates on informative subsets.

the average tokens per response. Unless noted, we keep all training and decoding hyper-parameters identical across baselines and our method, changing only the *response-level baseline* from the mean to a $K$-quantile (default $K=0.4$). This value is chosen to robustly balance exploration and exploitation; we present a detailed sensitivity analysis on $K$ in Appendix B.3.

## 5.1 OVERALL PERFORMANCE ACROSS MODELS & RECIPES

**Drop-in gains across model sizes.** Table 2 summarizes results on Qwen3-8B-Base and Qwen3-30B-A3B-Base. Replacing the mean baseline in DAPO with our K-quantile baseline (QAE) yields consistent pass@1 improvements across datasets and model sizes, while keeping pass@16 performance highly comparable. The stability of this process is further illustrated by the training dynamics curves for both 8B and 14B models in Appendix B.4, which show QAE consistently mitigates the entropy explosion seen in the baseline.

**Compatibility with strong recipes.** QAE is orthogonal to token-level controls (*e.g.,* CLIP-COV, KL-COV) and sequence-level optimization (GSPO). When layered on top of these methods, QAE consistently provides further gains without altering their hyper-parameters.

## 5.2 TRAINING DYNAMICS & ENTROPY SAFETY

**Pass@1 improves while pass@16 stays comparable.** Figure 5 (**Left**) plots AIME'24 performance over training for Qwen3-8B-Base. From ∼step 100, DAPO exhibits an entropy surge and *pass@1* stalls, while QAE maintains stable training and continues to improve. *Pass@16* remains similar,

**Negative-advantage entropy is the driver of instability.** Figure 5 (**Middle**) decomposes entropy by the sign of the advantage. The growth is dominated by *negative-advantage* samples; QAE suppresses this component and keeps the overall entropy within a productive range. This behavior follows directly from using a quantile baseline that down-weights uninformative negatives.

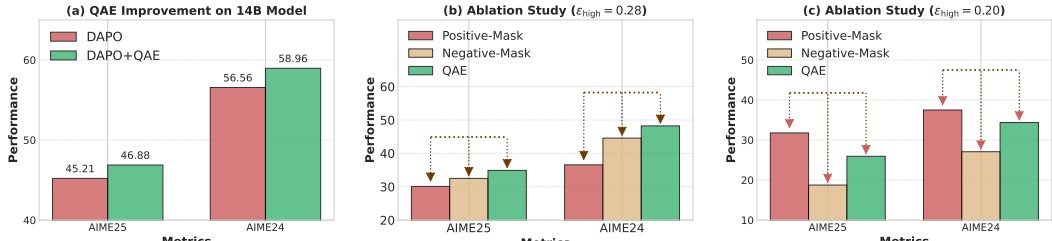

Figure 6: **Performance and ablations.** **(a)** QAE improves DAPO on the 14B model for both AIME'25 and AIME'24 (pass@1). **(b)** With weaker high-end clipping ($\epsilon_{\text{high}}$=0.28), controlling negative-advantage updates (NEG-MASK) is most critical, closely tracking full QAE. **(c)** With stronger clipping ($\epsilon_{\text{high}}$=0.20), positive-advantage control (POS-MASK) dominates.

**Response-level sparsity: the 80/20 rule.** Figure 5 (**Right**) shows that ≈80% of sampled responses have *zero* advantage throughout training. This "response-level 80/20 rule" focuses updates on the informative minority, explaining QAE's stability and efficiency. In contrast to the baseline, which leads to homogenized exploration (Sec. 3.2), QAE sustains a productive co-growth of diverse exploratory tokens and reasoning accuracy, as detailed in Appendix B.2.

## 5.3 ABLATIONS & COMPOSITION

**Masking mechanisms.** QAE can be viewed as selectively masking updates. To disentangle their roles, we define two one-sided objectives:

$$\mathcal{J}_{\text{POS-MASK}}(\theta) = \mathbb{E}_q\Big[\mathbf{1}_{\{p(q)\leq 1-K\}} \sqrt{\frac{p(q)}{1-p(q)}} \, \mathbb{E}_{o\sim\pi_{\text{old}}^+} s_\theta^+(o,q) - \sqrt{\frac{1-p(q)}{p(q)}} \, \mathbb{E}_{o'\sim\pi_{\text{old}}^-} s_\theta^-(o',q)\Big]. \quad (7)$$

$$\mathcal{J}_{\text{NEG-MASK}}(\theta) = \mathbb{E}_q\Big[\sqrt{\frac{p(q)}{1-p(q)}} \, \mathbb{E}_{o\sim\pi_{\text{old}}^+} s_\theta^+(o,q) - \mathbf{1}_{\{p(q)>1-K\}} \sqrt{\frac{1-p(q)}{p(q)}} \, \mathbb{E}_{o'\sim\pi_{\text{old}}^-} s_\theta^-(o',q)\Big]. \quad (8)$$

**Masking mechanisms.** QAE can be interpreted as masking *positives* on easy queries and *negatives* on hard queries. We isolate each side by constructing two objectives: POS-MASK (Eq. 7) and NEG-MASK (Eq. 8), leaving the other side unmasked.

**Explosion vs. collapse regimes.** As shown in Fig. 6 (**b-c**), when the high-end clipping is *weak* ($\epsilon_{\text{high}}$=0.28), the dominant failure mode is entropy explosion; NEG-MASK nearly matches QAE and outperforms POS-MASK. With *strong* clipping ($\epsilon_{\text{high}}$=0.20), collapse pressure dominates and the ordering flips (POS-MASK > NEG-MASK). This matches the two-regime analysis in Sec. 4.3.

## 6 DISCUSSION

$K$ **as an entropy–guided exploration–exploitation knob.** We use a single hyperparameter $K$ to control how many responses receive nonzero advantage, thereby steering the exploration–exploitation trade-off by modulating entropy. *Operational rule-of-thumb:* we select $K$ *once per baseline* by inspecting the entropy of the *baseline policy*, rather than the evaluation metric. When entropy is low (risk of mode collapse), choose $K = 0.6$ to inject diversity; when entropy is high (risk of unstable updates), choose $K = 0.4$ to temper exploration. Since all our recipes use `Clip-Higher`, we default to $K$=0.4; finer-grained tuning can yield further gains.

**QAE prioritizes *who* learns over *how much*.** Updating only a small subset of samples ($\sim 20\%$; Figure 5 (Right)) makes RLVR more stable and improves scaling behavior, indicating that *selection*, not update magnitude, is the primary bottleneck. QAE implements a binary reward with a quantile baseline at the *query* level: for difficult queries it assigns credit to successes, whereas for easy queries it assigns credit to failures. By adjusting the masking range—or by introducing dual masks—naïve DAPO/GRPO reduce to special cases, providing a safe fallback within the same framework.

**Baseline design as a third knob for entropy control.** Prior work has studied positive/negative ratios (Zhu et al., 2025), entropy dynamics (Cui et al., 2025), and advantage shaping (Cheng et al., 2025). QAE is orthogonal: it uses *baseline design*—a shift from the mean to the $K$-quantile—as the primary entropy lever and composes cleanly with existing techniques (Table 2). Related to Arnal et al. (2025), which analyzes tunable baselines in REINFORCE, our quantile baseline is a data-adaptive, group-level instantiation that improves robustness while preserving standard policy-gradient updates.

## 7 RELATED WORK

**Reinforcement learning for LLM** RL has become a key technique for eliciting advanced reasoning in large language models (LLMs), a paradigm shift from its earlier applications in preference alignment via RLHF (Ouyang et al., 2022). This modern approach, termed Reinforcement Learning with Verifiable Rewards (RLVR) (Lambert et al., 2024; Mroueh, 2025), leverages outcome-based optimization to achieve state-of-the-art performance in complex domains like mathematics and programming. Seminal works, including OpenAI's o1 (ope) and DeepSeek R1 (DeepSeek-AI et al., 2025), demonstrated that RL can effectively scale reasoning capabilities, spurring a new line of research (Yang et al., 2025a; Team et al., 2025). Central to this progress are online, value-free algorithms that have generally outperformed offline preference optimization methods (Rafailov et al., 2023; Wu et al., 2024; 2025). In particular, Group Relative Policy Optimization (GRPO) (Shao et al., 2024) and its successor, Dynamic Sampling Policy Optimization (DAPO) (Yu et al., 2025), have emerged as foundational baselines for many contemporary reasoning systems (Yue et al., 2025; Zeng et al., 2025; Hu et al., 2025). Our work uses DAPO as a representative algorithm to investigate a critical, unresolved challenge in this domain: the training instability caused by dysregulated policy entropy, which limits the performance and scalability of current RLVR methods.

**Exploration, entropy dynamics, and collapse/explosion in RLVR.** Existing RLVR entropy research follows three strands: (i) *Mechanistic* analyses identifying where exploration concentrates, such as high-entropy "forking" tokens (Wang et al., 2025b) or "thinking tokens" (Qian et al., 2025), and the dynamics of sequence-level collapse or explosion (Cui et al., 2025); (ii) *Objective-level* regulation steering entropy via modified optimization targets (Agarwal et al., 2025; Zhu et al., 2025; Zhang et al., 2025) or regularized MDP scheduling (Geist et al., 2019; Ahmed et al., 2019; Xu et al., 2021); and (iii) *Recipe/system-level* heuristics injecting exploration via advantage shaping (Cheng et al., 2025), Pass@k training (Chen et al., 2025), agentic scaffolds (Zhou et al., 2025; Shang et al., 2025), and modulated gradients (Wang et al., 2025a; Song et al., 2025). Despite these advances, a *baseline-level* entropy control that is data-adaptive yet preserves standard policy-gradient updates remains missing. Our work fills this gap using a quantile baseline and binary masking, offering a drop-in lever that complements existing methods while explicitly targeting stability.

## 8 CONCLUSION

**Conclusion** We propose *Quantile Advantage Estimation* (QAE), replacing the mean baseline with a group-wise $K$-quantile to implement a two-regime gate that amplifies rare successes and suppresses residual failures. Under first-order policy updates, QAE provides two-sided entropy control with bounded one-step entropy change, curbing both collapse and explosion. Empirically, QAE stabilizes entropy, sparsifies credit assignment, and improves pass@1 across reasoning benchmarks while composing cleanly with standard sequence- and token-level controls.

**Limitations and Future Work** (i) **Dynamic $K$:** Beyond a fixed $K$, explore simple schedules or two-phase curricula to better balance exploration and exploitation; (ii) **Automatic $K$:** Adapt $K$ to model state (*e.g.,* success rate, entropy, or gradient variance) to remove manual tuning; (iii) **PPO integration:** Embed the quantile-baseline idea into PPO's whitening/normalization—*e.g.,* batch-wise quantile baselines—to test robustness across algorithms and scales.

ACKNOWLEDGMENTS

This research is supported by the National Natural Science Foundation of China (U25A20445, 62572449, 62525211, 62302321). This research also benefited from the advanced computing resources provided by the Supercomputing Center of the USTC.

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

# A    PROOF

## A.1    PROOF OF PROPOSITION 4.1

**Proposition 4.1** (Quantile-regulated objective). *Assume binary rewards, group size $G \geq 2$, and the right-continuous empirical quantile. Using the standardized advantage in Eqs. 3–4, the learning objective is (up to a constant factor depending on $\varepsilon$) equivalent to*

$$\mathcal{J}_{\text{Quantile}}(\theta) = \mathbb{E}_q \Big[ \mathbf{1}\{p(q) \leq 1 - K\} \sqrt{\tfrac{p(q)}{1-p(q)}} \, \mathbb{E}_{o \sim \pi_{\text{old}}^+(\cdot|q)} s_\theta^+(o, q)$$
$$- \mathbf{1}\{p(q) > 1 - K\} \sqrt{\tfrac{1-p(q)}{p(q)}} \, \mathbb{E}_{o' \sim \pi_{\text{old}}^-(\cdot|q)} s_\theta^-(o', q) \Big]. \tag{6}$$

*Proof.* Write $p = p(q)$ for brevity. Recall the token-normalized surrogate

$$\mathcal{J}(\theta) = \mathbb{E}_q \mathbb{E}_{o \sim \pi_0(\cdot|q)} \frac{1}{|o|} \sum_{t=1}^{|o|} f\left( \frac{\pi_\theta(o_t \mid q, o_{<t})}{\pi_0(o_t \mid q, o_{<t})}, A(o \mid q) \right), \tag{9}$$

and the positive/negative homogeneous scaling of $f$ (the same convention as in the main text):

$$f(x, c) = \begin{cases} c \, f^+(x, 1), & c > 0, \\ |c| \big( -f^-(x, 1) \big), & c < 0, \end{cases} \qquad \Longleftrightarrow \qquad f(x, -c) = -c \, f^-(x, 1) \ (c > 0). \tag{10}$$

For the binary reward $r(o \mid q) \in \{0, 1\}$ and the group statistics $\mathbb{E}_{o \sim \pi_0(\cdot|q)} r(o \mid q) = p$ and $\text{Var}_{o \sim \pi_0(\cdot|q)} r(o \mid q) = p(1-p)$, the standardized advantage used in the paper takes the form

$$A(o \mid q) = \begin{cases} \sqrt{\dfrac{1-p}{p}}, & r(o \mid q) = 1, \\ -\sqrt{\dfrac{p}{1-p}}, & r(o \mid q) = 0. \end{cases} \tag{11}$$

Under the $K$-quantile baseline described in Section 4 (right-continuous), responses are masked asymmetrically by the regime of $p$:

$$\text{if } p \leq 1 - K: \quad A^+(q) = \frac{1}{\sqrt{p(1-p)}}, \quad A^-(q) = 0; \tag{12}$$

$$\text{if } p > 1 - K: \quad A^+(q) = 0, \quad A^-(q) = -\frac{1}{\sqrt{p(1-p)}}. \tag{13}$$

Equivalently, among $\{r = 1, r = 0\}$ only one label contributes in each regime.

Plug equation 12 into equation 9 and decompose over $r \in \{1, 0\}$ (writing $\pi_0^+(\cdot \mid q)$ and $\pi_0^-(\cdot \mid q)$ for $\pi_0(\cdot \mid q)$ conditioned on $r = 1$ and $r = 0$, respectively):

$$\mathcal{J}(\theta) = \mathbb{E}_q \Big[ \mathbf{1}\{p \leq 1 - K\} \, p \, \mathbb{E}_{o \sim \pi_0^+(\cdot|q)} \frac{1}{|o|} \sum_t f\left( \frac{\pi_\theta(o_t \mid q, o_{<t})}{\pi_0(o_t \mid q, o_{<t})}, \frac{1}{\sqrt{p(1-p)}} \right) \tag{14}$$
$$+ \mathbf{1}\{p > 1 - K\} \, (1-p) \, \mathbb{E}_{o \sim \pi_0^-(\cdot|q)} \frac{1}{|o|} \sum_t f\left( \frac{\pi_\theta(o_t \mid q, o_{<t})}{\pi_0(o_t \mid q, o_{<t})}, -\frac{1}{\sqrt{p(1-p)}} \right) \Big].$$

Apply the homogeneity equation 10 separately to the two terms in equation 14. For $p \leq 1 - K$ the scalar is positive, and for $p > 1 - K$ it is negative, hence

$$\mathcal{J}(\theta) = \mathbb{E}_q \Big[ \mathbf{1}\{p \leq 1 - K\} \sqrt{\tfrac{p}{1-p}} \, \mathbb{E}_{o \sim \pi_0^+(\cdot|q)} \frac{1}{|o|} \sum_t f^+\left( \frac{\pi_\theta(o_t \mid q, o_{<t})}{\pi_0(o_t \mid q, o_{<t})}, 1 \right) \tag{15}$$
$$- \mathbf{1}\{p > 1 - K\} \sqrt{\tfrac{1-p}{p}} \, \mathbb{E}_{o \sim \pi_0^-(\cdot|q)} \frac{1}{|o|} \sum_t f^-\left( \frac{\pi_\theta(o_t \mid q, o_{<t})}{\pi_0(o_t \mid q, o_{<t})}, 1 \right) \Big].$$

Equation 15 is the claimed quantile-regulated objective: compared with the symmetric GRPO/DAPO weight $\sqrt{p(1-p)}$, the quantile baseline (i) *masks* one side (positives on easy queries with $p > 1 - K$ or negatives on hard queries with $p \leq 1 - K$) and (ii) *re-weights* the active side by the asymmetric factors $\sqrt{p/(1-p)}$ or $\sqrt{(1-p)/p}$. This completes the proof.

**Instantiating $f$ for GRPO.** For GRPO we use

$$f^+(x,1) = \min\big(x, \text{clip}(x, 1-\epsilon, 1+\epsilon)\big) = \min(x, 1+\epsilon), \tag{16}$$

$$f^-(x,1) = \max\big(x, \text{clip}(x, 1-\epsilon, 1+\epsilon)\big) = \max(x, 1-\epsilon), \tag{17}$$

which can be plugged into equation 15 directly. □

## A.2 PROOF OF PROPOSITION 4.2

**Proposition 4.2** (Two-regime entropy safety of $K$-quantile). *Fix $q$ and a non-uniform $\pi(\cdot \mid q)$. Then:*

1. ***Low-success (explosion-proof).*** *If $p(q) \leq 1 - K$ so $b_K(q) = 0$, then for any baseline $b \in [0,1]$ (including the mean $b = p(q)$ or token-level clipping/KL that keep $b$ unchanged),*

$$\Delta H(q; b_K) \leq \Delta H(q; b).$$

2. ***High-success (collapse-proof).*** *If $p(q) > 1 - K$ so $b_K(q) = 1$, then for any $b \in [0,1]$,*

$$\Delta H(q; b_K) \geq \Delta H(q; b).$$

*Proof.* Fix $q$ and a non-uniform softmax policy $\pi(\cdot \mid q)$. For any baseline $b \in [0,1]$ and binary reward $r \in \{0,1\}$, write

$$A_b(y,q) = r(y,q) - b, \qquad F_q(b) := \text{Cov}_{y \sim \pi(\cdot|q)}\big(\log \pi(y \mid q), \ \pi(y \mid q)\,(r(y,q) - b)\big).$$

The entropy–covariance identity for softmax policies under first-order logit updates (adapted from Cui et al. (2025)) gives

$$\Delta H(q; b) \approx -\eta\, F_q(b), \qquad \eta > 0. \tag{18}$$

**Step 1: Baseline monotonicity.** By bilinearity of covariance,

$$F_q(b) = \text{Cov}_\pi\big(\log \pi, \ \pi r\big) - b\, \text{Cov}_\pi\big(\log \pi, \ \pi\big) =: F_q(0) - b\, C_q. \tag{19}$$

Let $U := \pi(Y \mid q)$ for $Y \sim \pi(\cdot \mid q)$. Then $C_q = \text{Cov}(\log U, U)$. Since $u \mapsto \log u$ and $u \mapsto u$ are strictly increasing on $(0,1]$, they are co-monotone; hence $\text{Cov}(\log U, U) > 0$ whenever $U$ is non-constant, i.e., whenever $\pi(\cdot \mid q)$ is non-uniform (see, *e.g.,* Chebyshev's sum / rearrangement inequality (Hardy et al., 1952)). Therefore $C_q > 0$ and equation 19 shows that $F_q(b)$ is strictly decreasing in $b$, so by equation 18 the entropy change $\Delta H(q; b)$ is strictly *increasing* in $b \in [0,1]$.

**Step 2: Two-regime extremality of the $K$-quantile baseline.** For Bernoulli rewards with success rate $p(q)$, the $K$-quantile baseline is

$$b_K(q) = \begin{cases} 0, & p(q) \leq 1 - K, \\ 1, & p(q) > 1 - K, \end{cases} \qquad \text{(Eq. 4)}.$$

Because $\Delta H(q; b)$ increases in $b$ (Step 1), we have, for any $b \in [0,1]$,

$$p(q) \leq 1 - K \ \Rightarrow\ b_K(q) = 0 = \min[0,1] \ \Rightarrow\ \Delta H(q; b_K) \leq \Delta H(q; b),$$

$$p(q) > 1 - K \ \Rightarrow\ b_K(q) = 1 = \max[0,1] \ \Rightarrow\ \Delta H(q; b_K) \geq \Delta H(q; b).$$

Strict inequalities hold whenever $\pi(\cdot \mid q)$ is non-uniform and $b \neq b_K(q)$. These are exactly Items (1) and (2) of Proposition 4.2.

This establishes the claimed *two-regime entropy safety*: in the low-success regime ($p \leq 1 - K$) the quantile choice $b_K = 0$ minimizes the entropy increment (explosion-proof), whereas in the high-success regime ($p > 1 - K$) the choice $b_K = 1$ maximizes it (collapse-proof). □

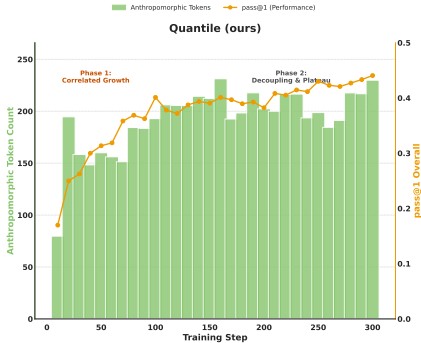

Figure 7: **High-entropy token diagnostics under QAE.** Green bars: counts of anthropomorphic high-entropy tokens; orange line: overall pass@1. Early coupled growth transitions to later decoupling—token counts plateau while accuracy improves—indicating entropy-safe, selective exploration.

## B  EXPERIMENTS

### B.1  IMPLEMENTATION DETAILS

**Experimental Setup:** Our configuration includes clip-higher, dynamic sampling, token-level policy gradient loss, and overlong reward shaping, as proposed in DAPO. We use the recommended hyperparameters: $\epsilon_{\text{high}} = 0.28$ and $\epsilon_{\text{low}} = 0.2$ for clip-higher, and a maximum response length of 20,480 with a 4,096-token cache for reward shaping.

**Training Details:** We train with a global batch size of 512, using 16 gradient accumulation steps with a mini-batch size of 32. The learning rate is fixed at $10^{-6}$ with no warmup or decay schedule. Importantly, we exclude both KL divergence and entropy losses.

**Evaluation:** To analyze scaling effects, we apply this method to the Qwen3-32B and Qwen3-8B base models, training them on the DAPO-Math-17K dataset (Yu et al., 2025).

**Additional Experiments:** We also conduct a cold-start experiment with the GSPO algorithm, initializing from the Qwen3-30B-A3B-Base model. In this configuration, we use four gradient accumulation steps per batch. The GSPO clipping ranges are set to $3 \times 10^{-4}$ (left) and $4 \times 10^{-4}$ (right), aligning with the official VERL implementation script[2].

### B.2  MORE EXPERIMENTS

**QAE sustains co-growth of "aha" markers and accuracy.**  Contrasting with `Clip-Higher`, Fig. 7 shows that under QAE the anthropomorphic token count *and* pass@1 rise together across training. From early to late steps, the green bars ("aha" markers) increase and remain elevated, while the orange curve improves monotonically, indicating that exploration is converted into productive reasoning rather than unchecked entropy.

**High-entropy token diagnostics under QAE (fine-grained snapshots).**  A finer-grained inspection at representative steps—**20/80/200** in Fig. 8—corroborates this interpretation. At step 20, anthropomorphic markers are sparse, consistent with exploration just being activated; by step 80, these tokens separate more distinctly, aligning with the performance uptick seen in the coupled-growth regime; by step 200, their counts stabilize despite continued pass@1 gains, evidencing a shift from "more randomness" to *targeted* refinement. Taken together with the trajectory view, these snapshots confirm that QAE leverages high-entropy branches when beneficial and then curbs their proliferation once they cease to deliver marginal utility.

---

[2]https://github.com/volcengine/verl/blob/main/recipe/gspo/test_gspo_3b_math.sh

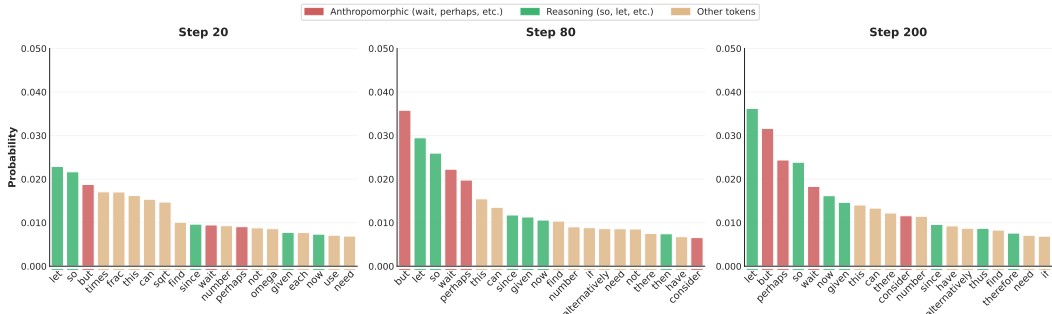

Figure 8: **Token-level diagnostics.** Probability mass over top high-entropy tokens at different training steps. Under QAE, exploratory tokens increase in a controlled manner, aligning with the stable-entropy regime in Fig. 5b.

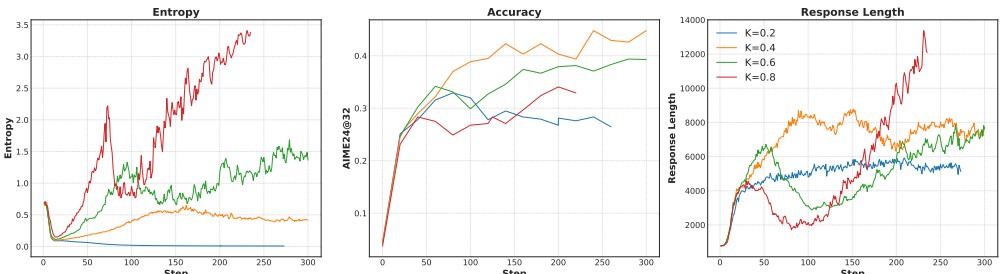

Figure 9: Training curves under different $K$ on Qwen3-8B-Base. Left: entropy; middle: accuracy (AIME24@32); right: response length.

### B.3 QUANTILE PARAMETER ANALYSIS

**Trade-offs governed by $K$.** Figure 9 (left/middle/right) shows how the quantile $K$ tunes entropy, accuracy, and response length for Qwen3-8B-Base. Large $K$ (e.g., $0.8$) marks most samples as negative-advantage, driving entropy upward, inflating response length, and yielding volatile training with an early accuracy plateau. Small $K$ (e.g., $0.2$) marks most samples as positive-advantage, producing a low-entropy, over-regularized regime that is stable but exploration-poor, with limited accuracy gains. These trends align with Sec. 4: $K$ simultaneously sets the share of responses updated and the direction of entropy flow.

**Stability at $K = 0.4$ (with `Clip-Higher`).** All main experiments use $K = 0.4$ with $\epsilon_{\text{high}} = 0.28$. This configuration avoids the high-entropy instability seen at $K = 0.8$ while maintaining sufficient stochasticity to prevent collapse. Empirically it yields bounded entropy (Fig. 9, left), stable lengths (right), and sustained accuracy improvements (middle), striking a robust exploration–exploitation balance and matching the two-sided entropy safety predicted by our analysis.

### B.4 ANALYSIS OF TRAINING DYNAMICS ON 8B AND 14B MODELS

**QAE stabilizes entropy and sustains performance gains across model scales.** We compare the baseline DAPO with DAPO+QAE on Qwen3-8B-Base (Figure 10) and Qwen3-14B-Base (Figure 11). Across both model sizes, QAE consistently reduces and stabilizes policy entropy, keeps response length bounded, and yields smoother, longer-lasting accuracy improvements.

On Qwen3-8B, the mean-baseline variant exhibits a pronounced entropy surge around step 100, accompanied by divergence in response length and a subsequent accuracy plateau. In contrast, QAE maintains entropy within a productive range throughout training and avoids the plateau, leading to sustained accuracy gains in later stages.

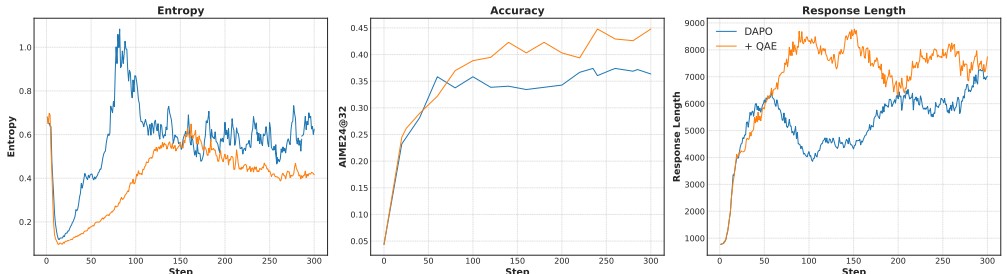

Figure 10: Training curves under DAPO and DAPO + QAE on Qwen3-8B-Base. Left: entropy; middle: accuracy (AIME24@32); right: response length.

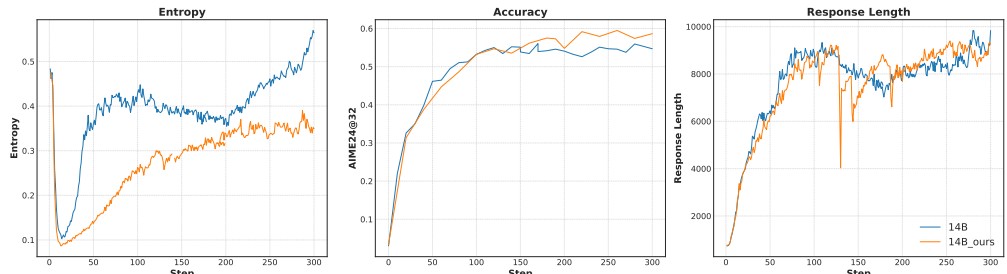

Figure 11: Training curves under DAPO and DAPO + QAE on Qwen3-14B-Base. Left: entropy; middle: accuracy (AIME24@32); right: response length.

The same pattern appears on Qwen3-14B. Although the entropy spike under the baseline is less severe, its entropy remains higher and more volatile than with QAE. QAE again moderates entropy and response length and produces a smoother, more monotonic accuracy trajectory. Taken together, these results indicate that QAE addresses the sensitivity of the mean baseline in value-free RL training and that principled baseline design provides an effective mechanism for scale-robust entropy control in RLVR.

