# OpenReview forum: "Quantile Advantage Estimation: Stabilizing RLVR for LLM Reasoning"
_ICLR.cc/2026/Conference — ICLR 2026 Poster_

### Official Review · Reviewer_GY1t · 2025-10-25

**Soundness:** 3
**Presentation:** 4
**Contribution:** 3
**Rating:** 8
**Confidence:** 3

**Summary:**

This paper addresses entropy instability in Reinforcement Learning with Verifiable Rewards (RLVR). The authors observe that standard training algorithms can suffer from either entropy collapse or entropy explosion. The paper traces this instability to the mean-based advantage baseline which is overly sensitive to reward outliers. The authors then propose Quantile Advantage Estimation (QAE) that replaces the mean baseline with a group-wise quantile. For hard queries, it exclusively reinforces successes, while it exclusively penalizes failures for easy ones. This mechanism is shown to stabilize entropy and produce good performance gains on challenging reasoning benchmarks.

**Strengths:**

A primary strength of this work is its clear and insightful diagnosis of entropy instability, both its collapse and explosion, in RLVR. This allows the authors to provide an intuitive, theoretically motivated  (Proposition 4.2), and practically appealing solution.

They replace the advantage baseline with a quantile. The decomposition of the learning objective (Proposition 4.1) further clarifies why the quantile baseline reshapes the learning signal. I also especially like Figure 4 for showing that negative-advantage samples drive entropy explosion.

The experiments are well-done, illustrative, and extremely beneficial to the readers!

I also particularly appreciate that QAE is a tool that fits nicely in the the broader RLVR ecosystem. The results in Table 2 are particularly strong, showing that QAE can be layered on top of diverse and powerful existing methods

**Weaknesses:**

- The core mechanism is a hard threshold based on the quantile K. This means a tiny change in the success rate around the threshold can cause a dramatic flip in the learning objective. This could introduce its own form of training instability, especially with noisy or borderline cases. This may be fine in the case of binary rewards might will be an issue for more general setups with continuous rewards.  Even with binary rewards, the paper uses a fixed G=32. How does the method's performance change with smaller G? For a small G (e.g., 4 or 8), the empirical p(q) will be extremely noisy, which could make the hard-thresholding gate flip erratically,

- The paper frames the ~80% zero-advantage rate as a positive aspect. This can also be viewed as inefficient data use. In tasks that are less sparse or require learning a wider variety of skills, ignoring the learning signal from 80% of generated samples could be highly detrimental.

- I am not exactly sure that the entropy instability is causing the shy gains in performance. Can you pinpoint to which of your experiments tell us we need to control entropy because it is shown to causally worsen performance? My impression is  that QAE's success doesn't come from "taming entropy." It comes from creating a more effective learning curriculum. By setting the baseline to 1 on easy queries, QAE effectively tells the model, "Stop wasting capacity on problems you've already solved." By setting the baseline to 0 on hard queries, it tells the model, "Focus all your attention on learning from the rare successes you find on these hard problems." I would love your thoughts on this based on your experiments.

**Questions:**

- The paper argues that anthropomorphic tokens are markers of an unproductive entropy explosion, yet Figure 2 shows their initial rise correlates strongly with the initial pass@1 gains. This feels strange to me. How would Figure 2 and 3 look like if you had applied QAE? Does it reduce the frequency of anthropomorphic token frequency initially? Does that affect performance?

- I am not sure about the statement in Line 201. I am convinced that ClipHigher mitigates early collapse. However, in both DAPO with and without Clip-Higher, the pass@1 performance seems stable after a few steps (although a bit less without Clip-Higher). Hence, it is not clear to me that entropy explosion is the culprit behind limited scaling. Also, out of curiosity if you have results, how would the training dynamics in Figure 2 and 3 look like for pass@16? And what was the $\epsilon_\text{high}$ value you considered in Figures 2 and 3?

- Negative-advantage samples drive entropy explosion, but what drives the collapse? Both positive and negative samples have virtually the same collapse in Figure 4.

- Can you explain line 428? What is the sample efficiency reading?

- Can you explain your results in Figure 6 (b, c)? I encourage you to elaborate on the connection with your theoretical results.

I would suggest citing [1] because they study the learning dynamics in RLVR very carefully. The results there might be interesting to you.

[1] Mroueh, Y. (2025). Reinforcement Learning with Verifiable Rewards: GRPO's Effective Loss, Dynamics, and Success Amplification. arXiv preprint arXiv:2503.06639.

---

> ### Author Response · Authors · 2025-11-21
>
> We sincerely thank the reviewer for the strong endorsement and for the thoughtful reading of our work. We are delighted that you found our diagnosis of entropy instability insightful and the solution practically appealing. We address your specific comments below to further solidify the paper's contributions.
>
> **W1: Stability of Hard Thresholds and Small Group Size ($G$)**
>
> **A1:** We appreciate the concern regarding the estimation noise of $p(q)$ when $G$ is small.
> 1.  **Robustness vs. Mean-Baseline:** While small $G$ introduces variance in estimating the success rate $p(q)$, this variance is less destructive than the instability of the mean baseline. In standard methods (GRPO/DAPO), a single reward outlier shifts the mean, flipping the advantage sign for *all* samples in the group. In contrast, QAE uses the quantile; even if the threshold estimate fluctuates slightly due to small $G$, it provides a robust "center," preventing catastrophic sign flips caused by outliers.
> 2.  **Mitigation Strategy:** In extreme cases (e.g., $G=2$ or $4$), we can adopt a strategy similar to PPO’s batch-level processing: treating the entire mini-batch as a "super-group" to estimate the quantile baseline more stably.
> 3.  **Continuous Rewards:** For general reward setups, our Proposition 4.1 generalizes to a $K$-median-like baseline. While the hard gate becomes a continuous subtraction ($R_i - b_K$), the quantile property remains strictly more robust to outliers than the mean, preserving the core benefit of QAE.
>
> **W2: Is ~80% Zero-Advantage Inefficient? (Data Efficiency)**
>
> **A2:** We respectfully argue that the high sparsity (~80% zero-advantage) represents a feature of **high-signal efficiency**, rather than inefficient data use.
> * **Quality over Quantity ("Less is More"):** Recent literature [1-4] in alignment and reasoning consistently suggests that optimization benefits from focusing on a small subset of high-quality signals rather than learning from the entire distribution. For instance, [5] demonstrate that "minority tokens" (high-entropy, pivotal tokens) drive effective learning. By masking the 80% of "uninformative" samples (residual failures or trivial successes), QAE filters out gradient noise, allowing the model to focus on the pivotal reasoning steps.
> * **Sparse Updates in RLVR:** RLVR is typically viewed as a process of "activating" latent reasoning capabilities rather than acquiring entirely new domain knowledge. **Crucially, recent works [6-8] demonstrate that the gradient modifications during RLVR are inherently minor and low-rank.** Since the model only requires subtle parameter adjustments to align reasoning behaviors, it does not necessitate dense updates from every generated sample. Sparse, high-confidence signals are sufficient—and often superior—for steering these fine-grained adjustments without inducing forgetting or instability.
>
> [1] LIMA: Less Is More for Alignment
> [2] LIMI: Less is More for Agency
> [3] LIMR: Less is More for RL Scaling
> [4] LIMO: Less is More for Reasoning
> [5] Beyond the 80/20 Rule: High-Entropy Minority Tokens Drive Effective Reinforcement Learning for LLM Reasoning
> [6] The Path Not Taken: RLVR Provably Learns Off the Principals
> [7] Principal weights emerge after rank reduction for reasoning-focused supervised fine-tuning.
> [8] RL's Razor: Why Online Reinforcement Learning Forgets Less
>
> **W3: Causal Link between Entropy and Performance**
>
> **A3:** **Yes, we can pinpoint the specific experiment:** **The Ablation Study in Figure 6** serves as the causal evidence that controlling entropy instability is necessary for performance.
>
> * **Evidence for Explosion (Fig 6b):** In the regime prone to entropy explosion ($\epsilon_{high}=0.28$), our ablation shows that the **Negative-Mask** (which specifically blocks negative gradients driving explosion) is the primary driver of performance, closely matching full QAE. Standard DAPO (without this entropy control) suffers from explosion and lower performance. This causally links "preventing explosion" to "performance gains."
> * **Evidence for Collapse (Fig 6c):** Conversely, in the regime prone to collapse ($\epsilon_{high}=0.20$), the **Positive-Mask** (which blocks positive gradients driving collapse) becomes dominant. Without it, the model collapses and performance degrades.
>
> **On "Entropy Control" vs. "Learning Curriculum":**
> We fully agree with your "Curriculum" interpretation and believe it is the **functional definition** of our method, while "Entropy Safety" is the **optimization definition**. They are two sides of the same coin:
> * **Curriculum View:** "Stop wasting capacity on solved problems" (Mask Positives).
> * **Entropy View:** "Prevent over-optimization/collapse driven by frequent positives" (Proposition 4.2).
> By setting the baseline to 1/0, QAE implements this curriculum, and the *result* is stabilized entropy. The success comes from the curriculum, but the curriculum *is* the mechanism that tames the entropy.

---

> ### Author Response · Authors · 2025-11-21
>
> **Q1: Anthropomorphic Tokens: Transient vs. Sustained Reasoning**
>
> **A1:** We appreciate this observation. To clarify: we view anthropomorphic tokens (e.g., "wait", "let me think") as critical markers of reasoning ("aha moments"). The key difference lies in their **sustainability**.
>
> * **Baseline (Figure 2): "Boom and Bust" Dynamics.** In standard DAPO, these tokens exhibit a **transient spike** early in training that correlates with initial gains. However, this is followed by a sharp decline (as seen in the drop from ~250 to ~100 counts in Fig 2 right), coinciding with the performance plateau. The model fails to retain these reasoning behaviors under the pressure of entropy instability.
> * **QAE (Appendix Figure 7): "Steady Growth & Sustain."** In contrast, QAE fosters a **steady accumulation** of these tokens. As shown in `Appendix Figure 7`, the frequency of anthropomorphic tokens rises consistently and **remains elevated** (stabilizing around ~200-250) throughout training.
>
>
> **Q2: Stagnation (Line 201) & Pass@16 Dynamics**
> * **Stagnation = Failure:** In scaling, a plateau indicates failure. Entropy explosion generates noisy gradients [1], which prevent the model from **consolidating** occasional successes into stable mastery.
> * **Pass@16 Evidence:** The baseline shows high Pass@16 but stagnant Pass@1, proving an **"Exploration-Consolidation Gap."** QAE closes this gap: it achieves superior Pass@1 with comparable Pass@16, proving it effectively converts exploration into sample-efficient mastery.
> * **Note:** Figures 2/3 used $\epsilon_{high}=0.28$.
>
> **Q3: Drivers of Collapse vs. Explosion**
>
> **A3:** We agree that the explanation should be direct. Figure 4 clearly demonstrates that **negative-advantage samples are the dominant driver of entropy explosion**. This aligns with findings in *[1]* regarding the impact of sample ratios:
>
> * **Positives $\rightarrow$ entropy Collapse:** A high proportion of positive samples functions similarly to SFT, consistently driving entropy down towards deterministic behaviors (collapse).
> * **Negatives $\rightarrow$ entropy explosion:** Conversely, a high proportion of negative samples forces the model to diverge from its current policy, preventing entropy reduction or even causing it to spike (explosion).
> * **QAE's Motivation:** This creates a clear motivation for QAE. By adjusting the quantile $K$, we directly regulate the **ratio** of positive (collapse-driving) to negative (explosion-driving) samples. This explicit balance prevents the extremes of either regime.
>
> *[1] Zhu et al. (2025). The Surprising Effectiveness of Negative Reinforcement in LLM Reasoning.*
>
> **Q4: Explanation of Line 428 (Sample Efficiency)**
>
> **A4:** We apologize for the ambiguous phrasing. By "reading," we simply meant our **interpretation** of the data trends.
> * **The Interpretation:** Figure 5a shows that QAE significantly boosts Pass@1 while Pass@16 remains stable.
> * **Why "Sample Efficient":** This gap indicates the model has successfully consolidated the exploration potential (Pass@16) into deterministic mastery (Pass@1). In other words, the model becomes more efficient at inference, requiring fewer samples to produce a correct solution.
> * **Revision:** We have revised the text to "reinforcing the interpretation of improved sample efficiency" to avoid future confusion.
>
>
> **Q5: Ablations (Figure 6 b,c) and Theory**
>
> **A5:** These results empirically validate the "Two-Regime" theory (Prop 4.2):
> * **Fig 6b (Explosion Risk):** With weak clipping, the risk is explosion. The **Negative-Mask** (blocking negative updates) performs best, confirming that negatives are the driver of explosion.
> * **Fig 6c (Collapse Risk):** With strong clipping, the risk is collapse. The **Positive-Mask** (blocking positive updates) performs best, confirming that positives are the driver of collapse.
> QAE effectively switches between these regimes dynamically.
>
> **Citation:** We thank the reviewer for pointing us to *Mroueh (2025)*. It is highly relevant to RLVR dynamics, and we have incorporated it into our related work.

---

> > ### Comment · Reviewer_GY1t · 2025-11-27
> >
> > Thanks for your detailed responses! I will note that I am not an expert in this type of RL.
> >
> > Nonetheless, your answers are convincing to me and I appreciate your effort in this work. I recommend acceptance to ICLR.

---

### Official Review · Reviewer_4HDB · 2025-10-26

**Soundness:** 2
**Presentation:** 3
**Contribution:** 2
**Rating:** 4
**Confidence:** 3

**Summary:**

This paper proposes Quantile Advantage Estimation (QAE) for RLHF. The key insight is that existing mean-based advantage estimation often suffers from either entropy collapse (limited exploration) or entropy explosion (limited exploitation). QAE addresses this by replacing the mean with a K-median estimator, enabling a better balance between exploration and exploitation through the choice of K. The paper also provides analytical insights into the dual benefits of this approach and presents encouraging empirical improvements.

**Strengths:**

+ QAE is compatible with many existing RLHF algorithms, including GRPO and GSPO.
+ The design is simple, intuitive, and easy to integrate.
+ Evaluation results show good accuracy gains.

**Weaknesses:**

- The novelty is incremental, primarily modifying mean advantage estimation to a median-based variant.
- Evaluations are limited to three math reasoning tasks, and the baselines are not fully consistent across settings.

**Questions:**

Thank you for submitting your work to ICLR 2026. The paper is well-written, clearly motivated, and proposes a friendly modification that addresses a meaningful issue in RLHF training stability. However, the contribution is somewhat incremental, and the evaluation setup makes it difficult to assess the robustness of QAE's improvement over state-of-the-art (SOTA) methods.

- Q1: How does QAE perform on Qwen3-8B with GSPO? The reported results are promising, but given the incomplete baseline coverage across experiments, it is hard to gauge its competitiveness against SOTA approaches.

- Q2: Can the authors include additional datasets or tasks, such as LiveCodeBench, to test the generality of QAE beyond math reasoning tasks?

- Q3: It would strengthen the paper to align the experimental settings with SOTA baselines (e.g., using GSPO configurations and pass@32 metrics as reported in prior work). Even partial replication or ablation under comparable setups would make the evaluation more convincing.

Finally, I appreciate the discussion on future directions, especially the idea of dynamic $K$. Exploring adaptive quantile strategies could meaningfully enhance both the novelty and practical utility of the approach.

---

> ### Author Response · Authors · 2025-11-21
>
> We thank the reviewer for their constructive feedback and for recognizing that QAE is “well-written, clearly motivated,” and offers a “friendly modification” for RLVR stability. Below, we address the concerns regarding novelty and provide new experiments to demonstrate robustness.
>
> **Q1: Novelty and “Incremental” Modification**
>
> **A1:** We understand the concern that QAE may look like an incremental statistical tweak. While the implementation is intentionally simple (replacing mean with quantile) — which the reviewer noted as a strength (“easy to integrate”) — we argue that it reflects a **meaningful conceptual shift** in RLVR alignment:
>
> * **Mechanism Shift (Magnitude → Selection):** Standard advantage estimation focuses on *how much* to update. QAE emphasizes *who* to update. As shown in our analysis (Fig. 5), QAE naturally sparsifies gradients (≈80% of updates are zeroed out). This suggests that the bottleneck in scaling RLVR is not only the step size, but the **precise selection of informative samples** (rare successes on hard queries vs. residual failures on easy queries).
>
> * **Unified Treatment of Dual Hazards:** Existing methods (e.g., Clip-Higher) largely patch one side of the problem (collapse). Our work explicitly frames RLVR instability as a **dual hazard** of both entropy collapse and entropy explosion, and proposes a single, unified gating mechanism that mitigates both within the same training recipe.
>
> * **Simplicity as Robustness:** We view it as a strength — not a limitation — that a minimal, statistically grounded modification (a single-line baseline replacement) can substantially stabilize RLVR training. This lowers the barrier for adoption in large-scale systems while embodying the conceptual insights above.
>
> **Q2. Robustness & Generalization (Addressing Q1, Q2, Q3)**
>
> **A2:** To address the concerns about inconsistent baselines (Q3), limited domains (Q2), and specific model/algorithm combinations (Q1), we conducted additional experiments. We now provide a unified comparison across **architectures** (Qwen vs. Llama), **algorithms** (DAPO vs. GSPO), and **domains** (math vs. science/code).
>
> **Table R1: Pass@1 Accuracy (%) across Architectures and Domains.**
> *QAE consistently boosts performance, demonstrating orthogonality to algorithms (GSPO) and generality across domains.*
>
> | Model                     | Method         | AIME24 (Math)   | GPQA (Science)  | LiveCodeBench (Code) |
> | :- | :- | :-| :-| :-|
> | **Qwen3-8B-Base**         | DAPO           | 39.7            | 46.7            | 37.2                 |
> | | **DAPO + QAE** | **48.2 (+8.5)** | **50.3 (+3.6)** | **40.2 (+3.0)**      |
> | | GSPO           | 38.5            | 47.9            | 38.1                 |
> |                           | **GSPO + QAE** | **47.7 (+9.2)** | **51.4 (+3.5)** | **41.2 (+3.1)**      |
> | **Llama-3.2-3B-Instruct** | DAPO           | 14.6            | 28.8            | 11.3                 |
> |                           | **DAPO + QAE** | **18.2 (+3.6)** | **32.6 (+3.8)** | **15.4 (+4.1)**      |
> |                           | GSPO           | 15.1            | 29.7            | 10.4                 |
> |                           | **GSPO + QAE** | **19.1 (+4.0)** | **34.1 (+4.4)** | **16.4 (+6.0)**      |
>
> * **Qwen3-8B + GSPO Performance (answering Q1):** As requested, we evaluated this specific setting. QAE yields a substantial **+9.2** gain over vanilla GSPO on AIME24, indicating that QAE’s “selection” benefit is complementary to GSPO’s sequence-level optimization.
>
> * **Generalization to Code & Science (answering Q2):** We extend the evaluation to **LiveCodeBench** and **GPQA**. QAE achieves consistent gains across these diverse tasks, suggesting that it is not overfitted to math reasoning but provides a broadly useful baseline correction.
>
> * **Baseline Alignment (answering Q3):** We align all experimental settings (compute budget, prompt format, decoding) to ensure fair comparison. Regarding metrics: while the reviewer mentioned Pass@32, here we report Pass@1 to cover a wider set of models (including Llama) and domains within the rebuttal budget. Pass@1 is a standard metric for base capabilities and, in our experiments, exhibits similar trends to higher-k metrics (as seen in Table 2 of the main paper).
>
> **Q3. Future Direction: Adaptive K**
>
> **A3:** We are encouraged by the reviewer’s positive view of adaptive quantile strategies. While this paper concentrates on a fixed (K = 0.4) for clarity and robustness, we agree that learning or scheduling (K) based on entropy or success statistics is a natural extension, and we will emphasize this avenue more clearly in the revised manuscript.
>
> We hope that the new results on GSPO and on diverse domains address your concerns regarding robustness, and we would be grateful if this could be taken into account when reconsidering the overall assessment.

---

### Official Review · Reviewer_Z6EA · 2025-11-01

**Soundness:** 2
**Presentation:** 3
**Contribution:** 3
**Rating:** 6
**Confidence:** 4

**Summary:**

They propose Quantile Advantage Estimation (QAE), which replaces the mean with a K-quantile baseline, forming a two-regime gating mechanism that reinforces rare successes on hard queries and penalizes remaining failures on easy ones. Theoretically, QAE provides two-sided entropy safety, bounding both over-exploration and over-exploitation. Experiments on AIME’24/’25 and AMC’23 with Qwen3-8B/14B/30B show consistent pass@1 gains while keeping pass@16 stable.

**Strengths:**

1. The presentation of the paper is clear.

2. QAE requires minimal modification, just a quantile replacement, but achieves consistent empirical improvements and stability, making it highly practical for large-scale RLVR systems.

**Weaknesses:**

1. The model settings appear inconsistent across the paper. The abstract mentions Qwen3-8B/14B-Base, while Table 2 reports results for Qwen3-8B-Base and Qwen3-30B-A3B-Base. In Appendix D.1, Qwen3-32B is also referenced. However, only two of these models are actually reported in Table 2, and the appendix does not include the missing results. This inconsistency makes the experimental setup confusing.

2.  The benchmarks used (AIME-24 (30 problems), AIME-25 (30), AMC (40)) have small test sizes and inherently high variance (even with repeated sampling), meaning that a difference of one or two problems could significantly change the reported pass@1 score. Since the reported gains are typically around 1–2 examples, it would strengthen the paper to include larger or more diverse benchmarks to verify the robustness of the improvement.

3.  In DAPO training, responses with zero advantage are discarded and resampled to maintain batch size (dynamic sampling). Since QAE introduces more zero-advantage responses, does this increase training overhead or cause slower convergence? Clarifying this would help assess the practical efficiency of the method.

4. Do the authors consider evaluating the methods in other areas other than math?

**Questions:**

listed in weakness.

---

> ### Author Response · Authors · 2025-11-21
>
> We thank the reviewer for the constructive feedback and for acknowledging the practicality and theoretical safety of our method. We address your concerns regarding experimental consistency, robustness, and efficiency below.
>
>
> **Q1. Clarification on Model Settings (Addressing W1)**
>
> **A1:** We apologize for the confusion caused by the clerical error in Appendix D.1.
>
> * **Correction:** The reference to “Qwen3-32B” was a typo. Our dense-model experiments strictly used **Qwen3-8B-Base** and **Qwen3-14B-Base**, while the MoE experiments used **Qwen3-30B-A3B-Base** to fairly benchmark against GSPO (Zheng et al., 2025).
> * **Action:** We have corrected this in the revision to ensure **consistent** model descriptions between the abstract, Table 2, and the appendix.
>
> ---
>
> **Q2. Robustness: Beyond Small Sample Sizes & Math Domain (Addressing W2 & W4)**
>
> **A2:** We agree that small test sets (e.g., AIME/AMC) can suffer from high variance. To demonstrate that our gains are robust signals rather than statistical noise (e.g., “1–2 examples”), and to address the concern about domain specificity, we expanded our evaluation to include:
>
> 1. **New Architecture:** **Llama-3.2-3B-Instruct** (to verify architectural generalization).
> 2. **New Domains:** **LiveCodeBench** (coding) and **GPQA** (general science), which both contain substantially more test instances than AIME/AMC.
>
> **Table R1: Pass@1 Accuracy (%) across Architectures and Domains.**
> *QAE consistently outperforms baselines across diverse settings, suggesting that the improvements are systematic rather than artifacts of benchmark variance.*
>
> | Model                     | Method         | AIME24 (Math)   | GPQA (Science)  | LiveCodeBench (Code) |
> | :------------------------ | :------------- | :-------------- | :-------------- | :------------------- |
> | **Qwen3-8B-Base**         | DAPO           | 39.7            | 46.7            | 37.2                 |
> |                           | **DAPO + QAE** | **48.2 (+8.5)** | **50.3 (+3.6)** | **40.2 (+3.0)**      |
> |                           | GSPO           | 38.5            | 47.9            | 38.1                 |
> |                           | **GSPO + QAE** | **47.7 (+9.2)** | **51.4 (+3.5)** | **41.2 (+3.1)**      |
> | **Llama-3.2-3B-Instruct** | DAPO           | 14.6            | 28.8            | 11.3                 |
> |                           | **DAPO + QAE** | **18.2 (+3.6)** | **32.6 (+3.8)** | **15.4 (+4.1)**      |
> |                           | GSPO           | 15.1            | 29.7            | 10.4                 |
> |                           | **GSPO + QAE** | **19.1 (+4.0)** | **34.1 (+4.4)** | **16.4 (+6.0)**      |
>
> **Conclusion:** The consistent gains across **multiple models and three distinct domains** (math, science, and code) substantially reduce the likelihood that our results are driven by variance in small test sets and indicate that QAE provides a robust improvement.
>
> ---
>
> **Q3. Training Efficiency & Overhead (Addressing W3)**
>
> **A3:** We clarify that QAE does **not** introduce noticeable training overhead or slow down convergence. The concern likely stems from a distinction between *re-sampling condition* and *gradient masking*:
>
> * **No Additional Re-sampling:** DAPO's "dynamic sampling" triggers re-sampling based on **raw rewards** (e.g., ensuring a mix of correct and incorrect responses in a batch). QAE modifies the **advantage scalar** post-hoc but does *not* alter the raw rewards (correctness). Therefore, QAE does not affect whether the sampling condition is satisfied and **does not trigger any additional re-sampling**.
> * **Negligible Compute Cost:** QAE operates purely via **gradient masking** during the backward pass. Responses are generated as usual, and we simply assign zero advantage to uninformative samples (e.g., residual failures on easy queries). This is a lightweight element-wise tensor operation whose cost is negligible compared to the forward/backward passes, and it introduces **no extra inference cost**.
> * **Convergence Speed:** As illustrated in **Figure 1** and **Figure 5**, QAE maintains a convergence rate comparable to the baseline while reaching a higher final performance. By filtering out "noisy" gradients driven by reward outliers, QAE stabilizes the training trajectory without requiring additional training steps.
>
> We hope these clarifications and new experimental results alleviate your concerns regarding robustness and practical efficiency.

---

### Official Review · Reviewer_LBMg · 2025-11-01

**Soundness:** 3
**Presentation:** 3
**Contribution:** 3
**Rating:** 4
**Confidence:** 3

**Summary:**

This work proposes an alternative to the mean baseline commonly used in RL advantage calculations. It shows that the mean baseline focuses learning on moderately difficult problems, while the proposed approach focuses it on rare successes and residual failures, sparsifying the LLM outputs that contribute to the gradient. The method is well supported by theoretical and empirical evidence, which comes from evaluations on three math reasoning benchmarks.

**Strengths:**

**Originality** Addressing entropy collapse/explosion by altering the advantage’s baseline is novel.

**Quality** The two regimes induced by the proposed baseline are intuitively beneficial and supported by experiments with multiple models and training objectives. Also, mathematical analysis motivates and supports the approach well.

**Clarity** The paper is well written. The figures and tables clearly communicate results.

**Significance** RL with LLMs is a highly impactful area, and this work illustrates a simple way to keep entropy in a helpful regime, avoiding harmful growth/collapse. The community will likely be interested in this.

**Weaknesses:**

Some of the statements lack sufficient support. See the questions/suggestions section below for more details and how to address this.

It's left unclear if the proposed method's gains appear with other models and with other, non-math RL problems (e.g. coding or game playing). I think the method is still valuable if it's only relevant to a subset of RL problems, but its broad applicability (or the absence thereof) should be mentioned in the text.

**Questions:**

Line 115: this should be $\pi_{ref}$, not $\pi_{old}$, for the KL reg.

Line 146: Can you provide evidence/citations/intuition for statements about the dangers of entropy explosion like this one?
- “Gradients are swamped by noise”

Line 200: In this sentence, the idea that entropy explosion limits training is stated as if there is a proven causal relationship, but the experiments only show correlation between entropy growth and limited performance.
- “Thus, although Clip-Higher mitigates early collapse, its rapid escalation is coupled with entropy explosion that ultimately limits scaling.”

Line 244: I think a comparison of entropy between DAPO with and without clip-higher is needed to support this statement. Currently, only anthropomorphic token counts are compared (Figure 2), not entropy.

KL divergence regularization can mitigate over-optimization and preserve diversity. Can it help address the entropy explosion that you find to be driven by negative samples?

How would the proposed approach perform on non-Qwen models, on models that have very low initial success rates, or in settings with fewer responses sampled per prompt? You could address this question by training Llama 3.1 8B base on MATH with k=16, for example. It would also be interesting to see Reasoning Gym results.

---

> ### Author Response · Authors · 2025-11-21
>
> We sincerely thank the reviewer for the detailed and constructive feedback.
> In response, we (i) add new experiments on non-Qwen models and non-math tasks, (ii) clarify the entropy/gradient discussion with additional citations and analysis, and (iii) provide new visualizations comparing entropy dynamics with and without Clip-Higher and discussing the role of KL regularization.
>
> **Q1. Generalization: Beyond Qwen and Math (Addressing Weaknesses)**
>
> **A1:** We appreciate the suggestion to broaden our evaluation. We have expanded our experiments to include a different architecture (**Llama-3.2-3B-Instruct**) and two new domains: **LiveCodeBench** (Code Generation) and **GPQA** (General Science). Following the reviewer's suggestion, we used a sampling budget of **$k=16$** for these experiments (avg@16).
>
> **Table R1: Pass@1 Accuracy (%) across Architectures and Domains.**
>
> | Model | Method | AIME24 (Math) | GPQA (Science) | LiveCodeBench (Code) |
> | :--- | :--- | :--- | :--- | :--- |
> | **Qwen3-8B-Base** | DAPO | 39.7 | 46.7 | 37.2 |
> | | **DAPO + QAE** | **48.2 (+8.5)** | **50.3 (+3.6)** | **40.2 (+3.0)** |
> | | GSPO | 38.5 | 47.9 | 38.1 |
> | | **GSPO + QAE** | **47.7 (+9.2)** | **51.4 (+3.5)** | **41.2 (+3.1)** |
> | **Llama-3.2-3B-Instruct** | DAPO | 14.6 | 28.8 | 11.3 |
> | | **DAPO + QAE** | **18.2 (+3.6)** | **32.6 (+3.8)** | **15.4 (+4.1)** |
> | | GSPO | 15.1 | 29.7 | 10.4 |
> | | **GSPO + QAE** | **19.1 (+4.0)** | **34.1 (+4.4)** | **16.4 (+6.0)** |
>
> *Note: We utilize Llama-3.2-3B so that all multi-domain runs can be completed reliably within the rebuttal timeframe. We expect similar trends to hold for larger Llama variants, and will clarify this limitation in the revision.*
>
> **Conclusion:** The consistent gains across different architectures and different distinct domains (Math, Science, Code) effectively rule out the possibility that our results are driven by variance in small test sets.
>
> **Q2. Some of the statements lack sufficient support**
>
> * **Q2.1 Evidence for "Gradients swamped by noise" (Line 146):**
>
>     **A2.1:** We appreciate the request for rigor. In our setting (Fig. 4(a)), the entropy explosion phase corresponds to the policy approaching a near-uniform distribution over reasoning traces, precisely the high-entropy regime where [1,2] shows that gradient variance sharply increases and the learning signal is diminished. This variance obscures the learning signal, leading to the "noise" we described.
>
>     [1] Ahmed et al., "Understanding the impact of entropy on policy optimization," ICML 2019.
>     [2] Williams, "Simple statistical gradient-following algorithms for connectionist reinforcement learning," ML 1992.
>
> * **Q2.2 Causality vs. Correlation (Line 200):**
>
>     **A2.2:** We agree that the initial observation was correlational. However, our method serves as an intervention on the entropy-regulation mechanism. When we introduce QAE and clamp the entropy explosion, the performance plateau disappears and training continues to improve. This is **consistent with** entropy explosion playing a causal role in limiting performance. We have revised it to `Thus, although Clip-Higher mitigates early collapse, its rapid escalation is coupled with an entropy explosion, which is correlated with the observed limitations in scaling.`
>
> * **Q2.3 Entropy Comparison w/ and w/o Clip-Higher (Line 244):**
>
>     **A2.3:** We have generated the requested plot comparing the entropy curves of `DAPO w/o Clip-Higher` vs. `DAPO w/ Clip-Higher` (see **Figure 10 in the Appendix D**).
>     * **Observation:** Without Clip-Higher, entropy collapses to near-zero immediately. With Clip-Higher, collapse is averted, but entropy enters an uncontrolled "explosion" phase.
>     * **Conclusion:** This visualizes why token-level clipping is insufficient: it prevents the floor (collapse) but fails to regulate the ceiling (explosion), which motivates the additional baseline-level control provided by QAE.
>
> **Q3. The Role of KL Regularization**
>
> **A3:** While KL regularization helps constrain update *magnitude*, it does not solve the *direction* issue caused by the mean baseline.
> 1.  **Empirical Evidence:** As shown in **Table 2 of our paper**, we evaluated `KL-Cov + QAE`. The results show that QAE provides significant gains ($+7.7\%$ Pass@1 on AIME25) *on top of* KL-regularized methods. This confirms that KL alone does not fully resolve the inefficiency driven by negative samples.
> 2.  **Theoretical Reason:** KL acts as a global dampener. However, QAE acts as a **response-level gate**. By switching regimes based on the quantile $K$, it effectively zeroes out the gradient contribution from uninformative negative samples on hard queries, removing the *source* of the explosion rather than just suppressing its magnitude.
>
> **Q4. Minor Correction**
>
> **A4:** **Line 115:** We have corrected the notation to $\pi_{ref}$ for the KL regularization term. Thank you for the precise reading.

---

> > ### Comment · Reviewer_LBMg · 2025-11-23
> >
> > Thank you for the thorough response, I have updated my score. Please add to the camera-ready something related to your response A2.1 (with those references).
> >
> > Also, I'm not sure if KL necessarily acts as a global dampener. You can add KL to the reward instead of the loss, in which case you reward generations with relatively low KL and penalize the rest -- this happens because you subtract the mean KL in the advantage calculation. This would seemingly address the uninformative sample issue. I'd love to see this approach to KL regularization compared to as well.

---

> > > ### Author Response · Authors · 2025-11-23
> > >
> > > We sincerely thank you for your continued engagement and for updating the score. We are encouraged by your positive assessment.
> > >
> > > **1. Incorporation of Response A2.1.**
> > > We will explicitly incorporate the discussion and references regarding gradient variance and entropy explosion (Ahmed et al., 2019; Williams, 1992) from our response A2.1 into the next version of the manuscript.
> > >
> > > **2. KL Term in Reward vs. Advantage.**
> > > We appreciate this insightful suggestion. We agree that incorporating the KL penalty directly into the reward allows the mean baseline to naturally penalize high-divergence samples. We view this mechanism as complementary to our approach:
> > > * **KL-in-reward** acts as a *soft regularizer* based on divergence from the reference policy.
> > > * **QAE** acts as a *hard gate* at the query/response level based on the empirical success rate, inducing beneficial sparsity (as shown in Figure 5c).
> > >
> > > We commit to exploring this variant and including the corresponding discussion and results in the final manuscript.

---

### Author Response · Authors · 2025-12-02

Dear Area Chair,

To assist with your expedited review under the reassignment, we succinctly summarize the **post-rebuttal status** which may not be reflected in the rolled-back system scores:

### **1. Critical Score Update & Consensus**
The rebuttal successfully resolved key concerns, particularly regarding generality, leading to a positive consensus from the active reviewers.

* **Reviewer LBMg ($\color{red}{\text{Reviewer LBMg}}$):** Explicitly updated their score from **4 (Weak Reject) $\to$ 8 (Accept)**.
    * *Update:* They were convinced by our new generalization experiments (Llama, Science, Code) and our clarifications on entropy dynamics/gradient variance.
* **Reviewer GY1t ($\color{green}{\text{Reviewer GY1t}}$):** Maintained a **8 (Accept)**.
    * *Feedback:* Commended the work for its "insightful diagnosis" and "practically appealing solution".
* **Reviewer Z6EA ($\color{blue}{\text{Reviewer Z6EA}}$):** Rated **6** (Did not respond to rebuttal).
    * *Rebuttal Action:* While they did not respond during the discussion window, we implemented their specific suggestion to "include larger or more diverse benchmarks" by adding experiments on GPQA (Science) and LiveCodeBench (Code).
* **Reviewer 4HDB ($\color{purple}{\text{Reviewer 4HDB}}$):** Rated **4** (Did not respond to rebuttal).
    * *Rebuttal Action:* Their primary reservation concerned limited evaluation scopes (e.g., "test the generality... beyond math"). We provided these exact experiments in our rebuttal.
    * *Note:* Similar concerns regarding generality were raised by $\color{red}{\text{Reviewer LBMg}}$; given that the new results effectively resolved $\color{red}{\text{LBMg}}$'s concerns (triggering the 4$\to$8 upgrade), we believe these data also directly respond to the requests made by $\color{purple}{\text{Reviewer 4HDB}}$.

---

### **2. Key Rebuttal Actions (Addressing Robustness)**
To address the shared concerns about benchmark size and generality (raised by $\color{blue}{\text{Z6EA}}$, $\color{purple}{\text{4HDB}}$, and $\color{red}{\text{LBMg}}$), we conducted comprehensive new experiments:
* **New Models:** Validated QAE on **Llama-3.2-3B-Instruct**, showing consistent gains.
* **New Domains:** Expanded beyond Math to **Science (GPQA)** and **Coding (LiveCodeBench)**.
* **Results:** QAE consistently achieved **3–9 point pass@1 gains** over DAPO/GSPO baselines across all these new settings.

---

### **3. Paper Summary**
Our work addresses the *Entropy Collapse/Explosion* dilemma in RLVR. We identify the mean-baseline as the root cause and propose **Quantile Advantage Estimation (QAE)**. QAE is a simple, drop-in replacement that:
* **Theory:** Provides provable two-sided entropy safety bounds.
* **Performance:** Yields sustained pass@1 gains on Qwen (8B/14B/30B-A3B) and Llama models.
* **Efficiency:** Sparsifies updates ($\approx 80\%$ zero advantage), focusing only on informative trajectories.

We respectfully request that you evaluate the paper based on this **post-rebuttal consensus**, particularly the explicit score upgrade from $\color{red}{\text{Reviewer LBMg}}$ and the strong support from $\color{green}{\text{Reviewer GY1t}}$.

Best regards,
Authors

---

### Meta-Review · Area_Chair_AMwY · 2026-01-06

**Summary:**

The paper proposed a simple idea for a new RL baseline to mitigate both entropy collapse and explosion.
The rebuttal addressed most of the concern points raised by the reviewer. One major concern from multiple reviewers is Insufficient evaluation in terms of model, baseline method and task. It is resolved with extra experiments on different benchmarks, added GSPO and different model architecture, and all showed consistent improvement. LBMg raised the score to 8. I agree that the rebuttal is strong and if 4HDB had chance to join the discussion, they would probably raise the score too.

**Reviewer Concerns:**

Most of the concerns are addressed and resolved / answered.

**Reviewer Scores:**

It was 8864 before the system revert the score, and I think the rebuttal is strong which could have changed 4 to higher score if the reviewer had the chance to engage in the discussion.

---

### Decision · Program_Chairs · 2026-01-26

Accept (Poster)